# Uptake of COVID-19 vaccines and associated factors among adults in Uganda: a cross-sectional survey

Rawlance Ndejjo ,[1] Nuole Chen ,[2] Steven N Kabwama ,[3] Alice Namale,[1] Solomon Tsebeni Wafula ,[1] Irene Wanyana,[4] Susan Kizito,[1] Suzanne N Kiwanuka,[5] William Sambisa,[6] Lily L Tsai ,[2] Rhoda K Wanyenze[1]

For numbered affiliations see end of article.

**Correspondence to**
Rawlance Ndejjo;
rndejjo@musph.ac.ug

## ABSTRACT

**Objective** COVID-19 pandemic remains one of the most significant public health challenges ever faced globally. Vaccines are key to ending the pandemic as well as minimise its consequences. This study determined the uptake of COVID-19 vaccines and associated factors among adults in Uganda.

**Design, setting and participants** We conducted a cross-sectional mobile phone survey among adults in Uganda.

**Main outcome variable** Participants reported their uptake of COVID-19 vaccines.

**Results** Of the participants contacted, 94% (1173) completed the survey. Overall, 49.7% had received COVID-19 vaccines with 19.2% having obtained a full dose and 30.5% an incomplete dose. Among the unvaccinated, 91.0% indicated intention to vaccinate. Major reasons for vaccine uptake were protection of self from COVID-19 (86.8%) and a high perceived risk of getting the virus (19.6%). On the other hand, non-uptake was related to vaccine unavailability (42.4%), lack of time (24.1%) and perceived safety (12.5%) and effectiveness concerns (6.9%). The factors associated with receiving COVID-19 vaccines were older age (≥65 years) (Adjusted Prevalence Ratio (APR)=1.32 (95% CI: 1.08 to 1.61)), secondary (APR=1.36 (95% CI: 1.12 to 1.65)) or tertiary education (APR=1.62 (95% CI: 1.31 to 2.00)) and health workers as a source of information on COVID-19 (APR=1.26 (95% CI: 1.10 to 1.45)). Also, reporting a medium-income (APR=1.24 (95% CI: 1.02 to 1.52)) and residence in Northern (APR=1.55, 95% CI: 1.18 to 2.02) and Central regions (APR=1.48, 95% CI: 1.16 to 1.89) were associated with vaccine uptake.

**Conclusions** Uptake of COVID-19 vaccines was moderate in this sample and was associated with older age, secondary and tertiary education, medium-income, region of residence and health workers as a source of COVID-19 information. Efforts are needed to increase access to vaccines and should use health workers as champions to enhance uptake.

## STRENGTHS AND LIMITATIONS OF THIS STUDY

⇒ This study had a high response rate with over 94% of the participants consenting to participate in the phone survey.

⇒ Results from the backchecking with the same individuals showed high consistency with the survey results.

⇒ Being a mobile phone survey, the study participants were not representative of the population as only those with a mobile phone could participate.

⇒ Reporting of vaccination status could have been subject to social desirability bias, which we minimised by reminding participants that the study was only for research purposes.

been registered globally.[1] In Africa, more than 11 million confirmed COVID-19 cases and 251 953 deaths were reported since the onset of the epidemic. Within the same period, Uganda recorded 163 905 cumulative COVID-19 cases and 3596 confirmed deaths.[1] In response and under the advice of the WHO, many countries at the beginning of the pandemic implemented non-pharmaceutical interventions (NPIs) that restricted movement such as lockdowns and curfews. Several governments both globally and in Africa also closed schools, places of worship, recreation centres and public places. Governments also promoted regular hand and respiratory hygiene, wearing of facemasks, ensuring physical and social distancing and working from home.[2] These public health and social measures significantly impacted the delivery of routine healthcare services, caused job losses, disrupted education and formal and informal trade and increased gender-based violence and mental health disorders.[3–5]

Vaccines as key pharmaceutical interventions to contain COVID-19 were adopted almost 1 year into the pandemic globally. Uganda recorded its first confirmed case of COVID-19 on 21 March 2020 and received

## INTRODUCTION

COVID-19 has resulted in significant morbidity and mortality globally and negatively disrupted multiple socioeconomic sectors. As of 31 March 2022, over 488 million confirmed cases and 6.1 million deaths had

its first batch of COVID-19 vaccines 1 year later in March 2021. At the start, vaccination targeted high-risk groups including health workers, teachers, security personnel, persons older than 50 years and those with comorbidities. Starting August 2021 when the country received more doses of vaccines, vaccination was opened up to all Ugandans aged 18 years and above. Vaccines were largely available through designated health facilities, outreaches and mobile vaccination service points. The Ministry of Health (MoH) ran media campaigns to mobilise communities for COVID-19 vaccination working hand in hand with local government structures. High vaccination coverage was critical for containment of the pandemic, re-opening of the economy and reversal of the negative socioeconomic impacts of the NPIs. However, the opening up of eligibility for vaccination was marred with negative information and fears of vaccine hesitancy. In order to develop critical strategies to achieve high vaccination coverage, there is need for an in-depth understanding of factors influencing the uptake of COVID-19 vaccination. This study, therefore, sought to gather and analyse data to determine the uptake of COVID-19 vaccines and associated factors among adults in Uganda.

## METHODS
### Study setting
This study was conducted in Uganda located in Eastern Africa. The country has 136 districts distributed in four administrative regions (Northern, Eastern, Central and Western) which were all involved in the study. As of 2020, Uganda had an estimated population of approximately 41.8 million people.[6] Having registered its first confirmed case of COVID-19 in March 2020, the country had by November 2021 experienced two waves of the disease. The first wave of the pandemic occurred from August 2020 to February 2021 of various non-Delta variants while the second wave happened from May to October 2021 fueled by the Delta variant.[1 7]

### Study design and population
This was a cross-sectional mobile phone survey conducted in November 2021 among a nationally constituted sample of adults. The study enrolled persons aged >18 years sampled from the country's four administrative regions: Central, Eastern, Northern and Western. We excluded persons who said they were ill and unable to participate in the interview.

### Sample size estimation
To enable tracking changes in adherence to NPIs following the introduction of vaccines, we used a previous sample of study respondents from an earlier survey[8] whose data were collected in March 2021.[8] The sample size for the previous survey was determined using the Leslie Kish formula for cross-sectional studies[9] considering the following assumptions: two-sided Z statistic corresponding to a 95% CI (1.96), NPI adherence level

of 50% since no other study had been conducted to show the composite level of adherence, a precision of 5% and a design effect of 2.5.[8] Considering a non-response rate of 10%, the total sample size estimate was 1056 people.

### Sampling strategy
We used the sample from an earlier survey,[8] which was constituted following quota sampling. Quotas were set on age, sex and location proportionate to national COVID-19 case distribution statistics as of February 2021.[10] The distribution of cases at the time was as follows: age: 18–35 years (51%), 36–55 years (37%), 56–65 years (8%), 65+ years (4%); sex: male (60%) and female (40%) and region: Central (55%) and 15% for each of Eastern, Western and Northern regions. With quotas in place, a simple randomly selected sample was obtained among the eligible population using a database of phone contacts provided by a registered research firm. In cases of replacement of previous participants due to unavailability or refusal to participate, a similar case distribution was followed during sampling of new contacts.

### Data collection
Data were collected using a structured survey questionnaire (online supplemental file 1), with mostly closed-ended questions, informed by a review of published literature.[8 11 12] The questionnaire was pretested among 20 people from the four regions of Uganda and relevant adjustments were made. The questionnaire was translated into nine major local languages spoken in Uganda, namely: *Ateso, Luganda, Lugbara, Lugisu, Luo, Lusoga, Ngakarimojong, Runyankole-Rukiga* and *Runyoro-Rutooro*. An independent group of translators validated the questionnaire translations and any discrepancies were addressed. The final survey instrument in each language was programmed in SurveyCTO software, incorporating appropriate routing, conditional logic and other controls and uploaded on handheld mobile tablets. Bench testing of the survey questionnaire was conducted, and adjustments made before actual data collection. Trained research assistants with a minimum of a diploma in a health-related field, fluent in the survey languages and with experience in mobile surveys conducted the interviews. Research assistants made phone calls from a designated place in Kampala to the respondents from whom they sought verbal informed consent after explaining to them what the study entailed and entered data into the tablets. The average interview time was 26 min. Respondents who preferred to defer the phone interviews due to busy schedules or other reasons received follow-up phone calls based on agreed-upon appointment times. Daily checks of the survey data were conducted to monitor quality and intervene early and appropriately, as well as ensure adherence to established quotas. A team of supervisors oversaw the work of the research assistants ensuring that questions were asked appropriately, and respondents were interviewed in the language they were most comfortable with. At the end of the interview period, we

conducted back checking of 10% of respondents to ascertain the quality of collected data.

## Data management and analysis strategy

During data collection, each research assistant examined, edited and cleaned their data daily before uploading it to the server. Data were encrypted and anonymised on the server and later downloaded and exported to Stata V.15.0 for further cleaning. Data analysis was conducted in Rstudio V.1.4.1106 (RStudio, PBC). Descriptive statistics have been provided in the form of means (SD) for continuous variables while categorical variables have been expressed as frequencies and percentages. Socioeconomic status was generated as an additive index from six variables on household ownership of television, computer, sofa set, refrigerator and cassette/CD/DVD player, and access to electricity. The socioeconomic status index was then divided into tertiles. The dependent variable was self-reported uptake of COVID-19 vaccines, which constituted those who reported receiving at least one dose of any WHO approved COVID-19 vaccines. We also determined the intention to uptake COVID-19 vaccines by asking unvaccinated respondents if they intended to receive the vaccine. The independent variables included sociodemographic characteristics (age, gender, employment status, education and occupation, place of residence (urban vs rural, region)) and source of information on COVID-19. To determine the factors associated with vaccination uptake, we ran multivariable modified Poisson regressions with robust error variance and presented prevalence ratios (PR) and corresponding 95% CIs. Only variables with a p value ≤0.2 at bivariate levels were included in the final model.

## Patient and public involvement

No patients or members of the public were involved in the study design, setting the research questions, interpretation or writing up of results, or reporting of the research.

## RESULTS

### Sociodemographic characteristics of participants

Of the 1249 respondents reached, a total of 1173 (94%) participants completed the survey. The mean age of respondents was 39.7 years (SD±14.2) and majority 717 (61.1%) were men. Half 606 (51.7%) of the study participants were from the Central region, 548 (46.8%) had an urban residence and 548 (46.7%) belonged to the lowest socioeconomic tertile. Nearly 4 in 10 (39%) respondents had only primary or no formal education (table 1).

### Uptake of COVID-19 vaccines and intention to vaccinate

Among all respondents, 225 (19.2%) reported receiving a full dose of the vaccine and 357 (30.5%) an incomplete dose. Slightly above 60% of the respondents 367 (63.2%) reportedly experienced side effects following vaccination mostly fever 147 (40.1%), fatigue 115 (31.3%) and headache 101 (27.5%). Among those who had not received

**Table 1** Sociodemographic characteristics of participants

| Characteristic | Number of participants (n=1173) | Percentage (%)* |
|---|---|---|
| Sex | | |
| Male | 717 | 61.1 |
| Female | 456 | 38.9 |
| Age group (years), mean age (SD) | 39.7 (±14.2) | |
| 18–35 | 553 | 47.1 |
| 36–55 | 439 | 37.4 |
| 56–64 | 92 | 7.8 |
| 65+ | 89 | 7.6 |
| Region of residence | | |
| North | 182 | 15.5 |
| East | 211 | 18 |
| Central | 606 | 51.7 |
| West | 174 | 14.8 |
| Residence | | |
| Urban | 548 | 46.8 |
| Rural | 417 | 35.6 |
| Semi-urban | 207 | 17.7 |
| Not stated | 1 | |
| Earnings per month ($) | | |
| <14 | 256 | 25.6 |
| 14–29 | 226 | 22.6 |
| 30–57 | 196 | 19.6 |
| 58–143 | 229 | 22.9 |
| >143 | 93 | 9.3 |
| Not stated | 173 | |
| Education level | | |
| None or incomplete primary | 265 | 23.2 |
| Primary (completed) | 180 | 15.7 |
| Secondary | 431 | 37.7 |
| Tertiary | 268 | 23.4 |
| Not stated | 27 | |
| Socioeconomic index | | |
| Low | 548 | 46.7 |
| Middle | 435 | 37.1 |
| Higher | 190 | 16.2 |
| Religion | | |
| Catholic | 384 | 33.3 |
| Anglican | 372 | 32.3 |
| Born Again (Pentecostal) | 147 | 12.8 |
| Muslim | 226 | 19.6 |
| Other religions | 24 | 2.1 |
| Not stated | 20 | |

Continued

**Table 1** Continued

| Characteristic | Number of participants (n=1173) | Percentage (%)* |
|---|---|---|
| Current occupation | | |
| Unemployed | 193 | 17.1 |
| Employed | 182 | 16.1 |
| Self employed | 355 | 31.4 |
| Casual labourer | 67 | 5.9 |
| Farmer | 334 | 29.5 |
| Not stated | 42 | |
| Current household size, mean (SD) | 5.6 (3.5) | |
| 5 or fewer | 653 | 55.7 |
| 6–10 | 430 | 36.7 |
| More than 10 | 90 | 7.7 |

*Percentages calculated do not include respondents who did not record responses (eg, 'Not stated' in the tables).

**Table 2** Vaccination uptake and intention to vaccinate among participants

| Variable | Count | Percentage (%)* |
|---|---|---|
| Vaccination uptake | (n=582) | |
| Full dose (two shots) | 225 | 19.2 |
| Incomplete dose | 357 | 30.5 |
| No vaccination | 590 | 50.3 |
| Experienced any side effects after first dose | | |
| No | 214 | 36.8 |
| Yes | 367 | 63.2 |
| Side effects reported | (n=367) | |
| Fever | 147 | 40.1 |
| Fatigue | 115 | 31.3 |
| Headache | 101 | 27.5 |
| Muscle soreness/pain | 95 | 25.9 |
| Injection site reaction | 88 | 24.0 |
| Others† | 38 | 10.4 |
| Vaccination intention (among unvaccinated) | n=590 | |
| Intend to vaccinate | 537 | 91.0 |
| Did not intend to vaccinate | 48 | 8.1 |
| Did not know | 5 | 0.8 |

*Percentages calculated do not include respondents who did not record responses for example, 'Not stated' in the tables.
†Allergic reaction, cough, body pain, dizziness, arrhythmias, body weakness, paralysis for a few days, erectile dysfunction for a few days.

a vaccine, 537 (91.8%) reported intention to vaccinate (table 2).

### Reasons for vaccine uptake/non-uptake and intention/unintention to vaccinate

The reasons for COVID-19 vaccine uptake and intention to vaccinate were similar with both categories of respondents mostly reporting the need to obtain protection from COVID-19 and having a high perceived risk of getting the virus. Over 40% of respondents who had not been vaccinated attributed it to vaccine unavailability 250 (42.4%) and below a quarter of respondents to not having time 142 (24.1%). The reasons for lack of intention to vaccinate were mainly related to safety 24 (50.0%) and effectiveness concerns 17 (35.4%) which were similarly reported for non-uptake of vaccines (table 3).

### Willingness to vaccinate for different vaccine types

All respondents were asked if they would receive the different types of COVID-19 vaccines if offered at that point and were free of charge. Only 316 (26.9%) reported that they would take any vaccine regardless of the type and 488 (41.6%) indicated a willingness to take at least one type of the vaccine. The most preferred COVID-19 vaccine types were Johnson and Johnson 436 (37.4%) and AstraZeneca 405 (34.7%) (figure 1).

### Factors associated with uptake of COVID-19 vaccines

At the multivariable analysis level, participants aged >65 years had a 32% higher likelihood to have been vaccinated compared with those aged 18–35 years (adjusted PR=1.32, 95% CI: 1.08 to 1.61, p=0.008). Participants from the Northern (adjusted PR=1.55, 95% CI: 1.18 to 2.02, p=0.002) and Central regions (adjusted PR=1.48, 95% CI: 1.16 to 1.89, p=0.002), respectively had a 55% and 48% higher likelihood to have received the vaccine compared with those from the Western region. Participants with secondary (adjusted PR=1.36, 95% CI: 1.12 to 1.65, p=0.002) or tertiary education (adjusted PR=1.62, 95% CI: 1.31 to 2.00, p<0.001) were more likely to have received the COVID-19 vaccine compared with those with incomplete primary/no formal education. Respondents whose monthly income was between $30 and $57 (APR=1.24 (95% CI: 1.02 to 1.52), p=0.029) had a higher uptake of COVID-19 vaccines than those who earned <$14. Having health workers as a source of information on COVID-19 was associated with higher uptake of COVID-19 vaccines in Uganda (adjusted PR=1.26, 95% CI: 1.10 to 1.45, p=0.001) (table 4).

### DISCUSSION

This study examined the uptake of COVID-19 vaccines and associated factors among adults aged 18 years and above in Uganda. Among the study participants, about one in five (19.2%) reported receiving a full dose of the COVID-19 vaccine while 30.5% had received an incomplete dose. Over 90% of those who were unvaccinated reported the

**Table 3** Reasons for (non) uptake of COVID-19 vaccines and intention to vaccinate (multiple response)

| Reasons | Uptake of vaccines, n=582 (%) | Intention to vaccinate, n=537 (%) |
|---|---|---|
| To protect self from COVID-19 | 505 (86.8) | 458 (85.3) |
| High perceived risk of getting COVID-19 | 114 (19.6) | 90 (16.8) |
| Prioritised due to health (comorbidities) | 95 (16.3) | 34 (6.3) |
| Recommendation from health workers | 81 (13.9) | 38 (7.1) |
| Prioritised due to occupation | 74 (12.7) | – |
| Travel purposes | 44 (7.6) | 45 (8.4) |
| Job requirement | – | 82 (15.3) |
| Others | 21 (3.6)* | 20 (3.8)† |
| **Reasons** | **Non uptake of vaccines. n=590 (%)** | **No intention to vaccinate, n=48 (%)** |
| Vaccines are unavailable | 250 (42.4) | 1 (2.1) |
| Don't have time | 142 (24.1) | 2 (4.2) |
| Safety concerns | 74 (12.5) | 24 (50.0) |
| Doubt vaccine effectiveness | 41 (6.9) | 17 (35.4) |
| Not among eligible group | 30 (5.1) | 4 (8.3) |
| Transport costs | 24 (4.1) | Not reported |
| Don't know where to access the vaccines from | 20 (3.4) | Not reported |
| Do not fear COVID-19/ trust immunity | 10 (1.7) | 1 (2.1) |
| Others | 82 (14.0)‡ | 7 (14.6)§ |

*Requirement for school attendance, being exemplary, following MoH guidelines, boosting immunity, to access services, among the eligible group.
†Access to health services, government mandate, pressure from peers, to be exemplary, requirement for school attendance.
‡Pregnant, breastfeeding, waiting for another vaccine type, lack identification documents, long queues, currently sick, recently recovered from COVID-19.
§Religious beliefs, do not believe COVID-19, HIV positive and fear side effects, underlying hepatitis B infection, body already weak, lack of identification documents.
MoH, Ministry of Health.

intention to be vaccinated. The major reasons for vaccine uptake and intention to vaccinate were protection of self from COVID-19 and a high perceived risk of getting the virus while reasons for vaccine non-uptake were vaccine unavailability, the lack of time to go get vaccinated, and safety and effectiveness concerns. The factors that were associated with receiving the COVID-19 vaccine were older age (65 years and above), having secondary education and above, having a moderate income and reporting

health workers as a source of information on COVID-19. Being a resident of Northern and Central Uganda was also associated with a high likelihood of receiving the vaccine.

Uptake of COVID-19 vaccines in this sample of respondents was higher for both full and incomplete doses than the vaccinated proportion of the population as of November 2021 when this study was conducted. MoH data of 8 November 2021 indicated that 55.8% and 16.8% of the priority groups and 12.2% and 3.7% of the adult population had received their first and second doses of the vaccine respectively.[13] The higher-than-baseline vaccination coverage could be attributed to the use of mobile phones for the survey and thus the relatively urbanised study sample whose access to vaccines was higher than those in rural areas. Moreover, a high proportion of participants were from the Central region, which was most impacted by COVID-19, and their experiences could have influenced vaccine uptake. In addition, intention to vaccinate was very high at over 90%; higher than the combined 'definite intention' of 57.8% and 'probable intention' of 26.2% from the March survey round.[8] In a November 2021 survey among 23 000 respondents from 19 African Union members states including Uganda, (78%) of respondents had either been vaccinated or were likely to get vaccinated.[14] The second COVID-19 wave fueled by the Delta variant that was experienced in Uganda from June to September 2021 and led to at least 2800 deaths compared with the less than 300 recorded at the end of the first wave[1 15] could also have contributed to the high uptake of the vaccine and intention-to-vaccinate. In addition, there was concern about potential vaccine mandates including anticipation that the unvaccinated would be denied health and social services which could also have increased the intention to vaccinate.

The major reasons for vaccine uptake and intention to vaccinate were protection of self from COVID-19 and a high perceived risk of getting the virus, similar to previous research.[11] This is also an indication of the respondents' appreciation of the role of vaccines in preventing morbidity and saving lives. Those unvaccinated attributed it to vaccine unavailability and the lack of time. The survey in 19 African countries concluded that low vaccine uptake was mostly due to unpredictable supply of vaccines and logistical hurdles than reluctance or refusal to get vaccinated.[14] Earlier surveys in Uganda conducted in 2020 had also shown a high acceptance of COVID-19 vaccines of over 85%.[11 16] To bridge the willingness-intention-uptake gap in Uganda, the MoH should increase access and availability of COVID-19 vaccines. Evidence shows that strategies that take vaccines closer to the communities are likely to mitigate time and transport-related barriers and increase vaccine uptake.[17 18] This could be achieved by increasing the number of health facilities offering the vaccines, conducting more vaccination outreaches, or setting up mobile vaccine points. The WHO guidance has also emphasised the importance of location and time in COVID-19 vaccine uptake.[19] On the other hand, the

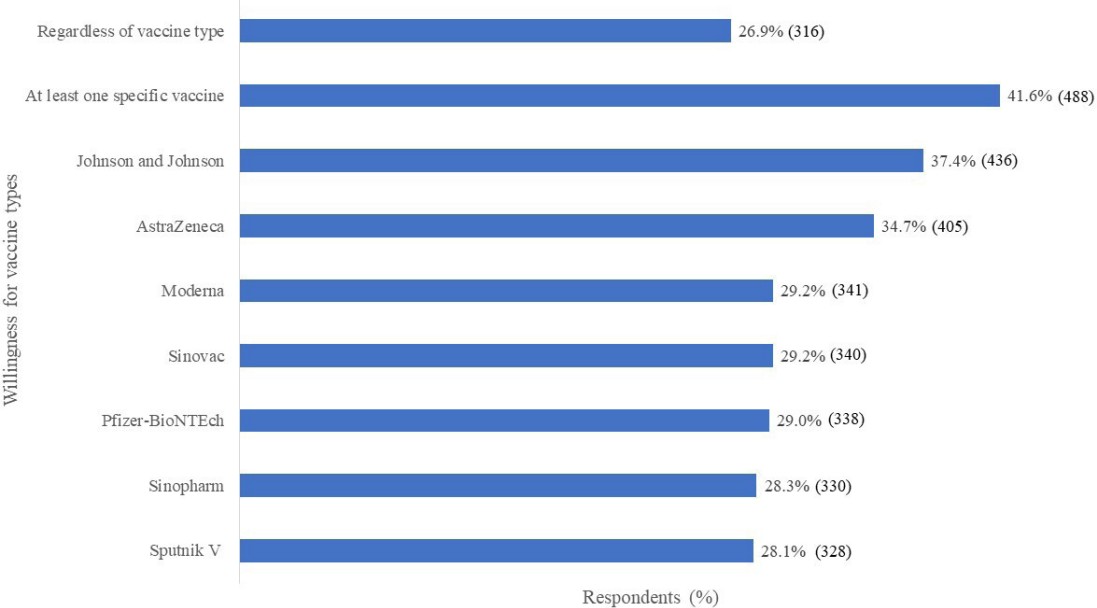

**Figure 1** Willingness for COVID-19 vaccination for different vaccine types.

study reported that safety and effectiveness concerns hindered vaccine uptake and intention to vaccinate similar to previous research.[8 11 20–23] Of note as well was the observed high prevalence (63%) of self-reported vaccine side effects which could go a long way in reinforcing safety concerns among the population. Vaccine adverse events should be monitored closely, and appropriate information, education and communication material developed including information on expected side effects to counter their potential effect on the uptake of vaccination by the unvaccinated. Accurate, consistent and transparent communication and dialogue about uncertainty, risks and anticipated benefits can go a long way in building confidence and trust in the COVID-19 vaccines and create motivation for vaccination.[19 23] This communication could also bridge observed gaps in vaccine preference to prevent vaccine adverse events from being a barrier to vaccination. The Johnson and Johnson vaccine being a single shot had a higher preference among respondents due to the perceived inconvenience and unpredictability of obtaining a second vaccine dose.

It was not surprising that those aged 65 years and above had a higher vaccination uptake as these were part of the prioritised group for COVID-19 vaccination in the country. Education status also predicted vaccination status similar to previous research on COVID-19 vaccine acceptability.[24–27] However, further efforts are required to ensure the dissemination of accurate and simple COVID-19 vaccination messages to those of lower education levels including translating information in the local languages so that this group is not left behind. A moderate income was associated with higher vaccine uptake; however, this relationship was not sustained with increasing income levels. The regional differences observed in the uptake of COVID-19 vaccines may have been due to differences in vaccine access and availability, especially for Central region which was most hit by the pandemic and was prioritised early during vaccine rollout. From previous research, income levels and locations have been reported as predictors of COVID-19 vaccine acceptability.[27 28]

One major finding from our work was that respondents whose source of information on COVID-19 was health workers had a higher likelihood for COVID-19 vaccination. This positions health workers as a key resource in increasing vaccination uptake, and thus they should be furnished with sufficient and accurate information and supported with effective communication tools to influence their clients at facility and community level. Previous studies report that health worker advice on vaccination was most trusted.[11 22] Health workers can lead health education and awareness programmes on COVID-19 and use their platforms at health facility and community level to influence the masses to uptake COVID-19 vaccines. However, vaccine uptake among health workers themselves was low at the time even when they were prioritised for vaccination from the start of the campaigns in Uganda and elsewhere. In a March 2021 survey in Uganda, just after the launch of the COVID-19 vaccination exercise, a vaccine acceptability rate of 37.3% and hesitancy of 30.7% were reported among medical students.[12] In a June to August 2021 online survey, acceptance or willingness to uptake the COVID-19 vaccine stood at over 97% and 65.3% of eye healthcare workers had received a COVID-19 vaccine shot influenced by high perceived susceptibility and benefits.[29] An in-depth study among health workers reported the lack of trust in the vaccine, fear of side effects, not feeling at risk, lack of sufficient information about vaccines, health systems challenges and religious beliefs as barriers to COVID-19 vaccination.[30] When health workers are vaccinated, they are more likely to recommend the same to their clients.[31] Therefore, appropriate interventions should be instituted to effectively

**Table 4** Factors associated with COVID-19 vaccine uptake among adults

| Variables/characteristics | Self-reported uptake of COVID-19 vaccine | | Unadjusted PR (95% CI)* | P value | Adjusted PR (95% CI)† | P value |
|---|---|---|---|---|---|---|
| | No (%) | Yes (%) | | | | |
| **Age in years** | | | | | | |
| 18–35 | 296 (53.5) | 257 (46.5) | 1 | | 1 | |
| 36–55 | 213 (48.6) | 225 (51.4) | 1.11 (0.97 to 1.26) | 0.124 | 1.09 (0.95 to 1.25) | 0.244 |
| 56–64 | 44 (47.8) | 48 (52.2) | 1.12 (0.91 to 1.39) | 0.292 | 1.17 (0.92 to 1.48) | 0.193 |
| 65+ | 37 (41.6) | 52 (58.4) | 1.26 (1.03 to 1.53) | **0.023** | 1.32 (1.08 to 1.61) | **0.008** |
| **Region of residence** | | | | | | |
| Western | 115 (66.5) | 58 (33.5) | 1 | | 1 | |
| Northern | 74 (40.7) | 108 (59.3) | 1.77 (1.39 to 2.25) | **<0.001** | 1.55 (1.18 to 2.02) | **0.002** |
| Eastern | 112 (53.1) | 99 (46.9) | 1.40 (1.09 to 1.80) | **0.010** | 1.29 (0.99 to 1.69) | 0.064 |
| Central | 289 (47.7) | 317 (52.3) | 1.56 (1.25 to 1.95) | **0.001** | 1.48 (1.16 to 1.89) | **0.002** |
| **Residence** | | | | | | |
| Urban | 270 (49.4) | 277 (50.6) | | | 1 | |
| Rural | 206 (49.4) | 211 (50.6) | 0.99 (0.88 to 1.13) | 0.990 | 1.11 (0.97 to 1.28) | 0.137 |
| Semi-urban | 114 (55.1) | 93 (44.9) | 0.89 (0.75 to 1.05) | 0.173 | 0.92 (0.75 to 1.11) | 0.373 |
| **Gender** | | | | | | |
| Male | 351 (49.0) | 366 (51.0) | 1 | | 1 | |
| Female | 239 (52.5) | 216 (47.5) | 0.93 (0.82 to 1.05) | 0.237 | 1.00 (0.87 to 1.14) | 0.973 |
| **Wealth index** | | | | | | |
| Low | 290 (53.0) | 257 (47.0) | 1 | | 1 | |
| Middle | 217 (49.9) | 218 (50.1) | 1.07 (0.94 to 1.21) | 0.328 | 1.06 (0.91 to 1.24) | 0.442 |
| High | 83 (43.7) | 107 (56.3) | 1.20 (1.03 to 1.40) | **0.021** | 1.03 (0.83 to 1.28) | 0.758 |
| **Current occupation** | | | | | | |
| Unemployed | 91 (47.2) | 102 (52.8) | 1 | | 1 | |
| Employed | 76 (41.8) | 106 (58.2) | 1.10 (0.92 to 1.32) | 0.294 | 1.03 (0.84 to 1.27) | 0.763 |
| Self employed | 196 (55.2) | 159 (44.8) | 0.85 (0.71 to 1.01) | 0.066 | 0.84 (0.68 to 1.02) | 0.078 |
| Casual labourer | 45 (67.2) | 22 (32.8) | 0.62 (0.43 to 0.90) | **0.011** | 0.73 (0.48 to 1.11) | 0.146 |
| Farmer | 164 (49.1) | 170 (50.9) | 0.96 (0.81 to 1.14) | 0.664 | 0.99 (0.82 to 1.19) | 0.931 |
| **Education level** | | | | | | |
| No formal education/ incomplete primary | 161 (60.8) | 104 (39.2) | 1 | | 1 | |
| Complete primary | 109 (60.6) | 71 (39.4) | 1.01 (9.79 to 1.27) | 0.966 | 1.00 (0.78 to 1.28) | 0.998 |
| Secondary education | 207 (48.0) | 224 (52.0) | 1.32 (1.11 to 1.58) | **0.002** | 1.36 (1.12 to 1.65) | **0.002** |
| Tertiary | 98 (36.6) | 170 (63.4) | 1.62 (1.36 to 1.93) | **<0.001** | 1.62 (1.31 to 2.00) | **<0.001** |
| **Household size (mean)** | 5.41 | 5.99 | 1.02 (1.01 to 1.03) | **<0.001** | 1.02 (1.00 to 1.03) | 0.071 |
| **Monthly income ($)** | | | | | | |
| <14 | 144 (56.2) | 112 (43.8) | 1 | | 1 | |
| 14–29 | 117 (51.8) | 109 (48.2) | 1.10 (0.91 to 1.34) | 0.324 | 1.08 (0.89 to 1.32) | 0.423 |
| 30–57 | 86 (43.9) | 110 (56.1) | 1.28 (1.07 to 1.55) | **0.009** | 1.24 (1.02 to 1.52) | **0.029** |
| 58–143 | 114 (49.8) | 115 (50.2) | 1.15 (0.95 to 1.39) | 0.154 | 0.98 (0.79 to 1.22) | 0.876 |
| >143 | 36 (38.7) | 57 (61.3) | 1.40 (1.13 to 1.73) | **0.002** | 1.16 (0.91 to 1.49) | 0.219 |
| **Health workers as source of information on COVID-19‡** | | | | | | |
| No | 245 (57.9) | 178 (42.1) | 1 | | 1 | |

Continued

**Table 4** Continued

| Variables/characteristics | Self-reported uptake of COVID-19 vaccine | | Unadjusted PR (95% CI)* | P value | Adjusted PR (95% CI)† | P value |
|---|---|---|---|---|---|---|
| | No (%) | Yes (%) | | | | |
| Yes | 345 (46.1) | 404 (53.9) | 1.28 (1.13 to 1.45) | <0.001 | 1.26 (1.10 to 1.45) | 0.001 |

*Bivariate analysis.
†Multivariable analysis.
‡Other sources of information included family members, friends/peers, radio, television, community members and social media among others which were dichotomised and included in the analysis but were not significant.

deal with vaccine hesitancy among health workers and have them as champions for COVID-19 vaccination.

### Study limitations and strengths

Being a mobile phone survey, the study participants were not representative of the population and only those with a mobile phone could participate, contributing to selection bias. However mobile phone coverage in Uganda has increased over the years; according to the Uganda National Household Survey 2020, 74.0% of Ugandans own mobile phones.[32] There was also potential for social desirability bias, especially regarding reporting vaccination status which we minimised by reminding participants that the study was only for research purposes. Also, as a cross-sectional survey, the direction of associations observed is not clear. On the other hand, our study had a high response rate with over 94% of the participants consenting to participate. The high response rate could be attributed to following up previous survey participants, flexibility in conducting interviews at convenient times, as well as the time compensation (phone credit of 1.5 US dollars) provided. Results from the backchecking with the same individuals also showed high consistency with the survey results. Our study provides insights into COVID-19 vaccination uptake and intention to vaccinate which can facilitate the development of context-relevant strategies to increase vaccinations.

### CONCLUSIONS

Half of the study respondents were vaccinated against COVID-19, which was associated with older age, higher education level, moderate income, region of residence and reporting health workers as the source of COVID-19 information. Among the unvaccinated, over 90% expressed intention to vaccinate. Efforts are needed to increase access to vaccines and use health workers as a key resource in sharing information and champions to influence the masses which should positively impact uptake of COVID-19 vaccines.

**Author affiliations**
[1]Department of Disease Control and Environmental Health, School of Public Health, College of Health Sciences, Makerere University, Kampala, Uganda
[2]Department of Political Science, Massachusetts Institute of Technology, Cambridge, Massachusetts, USA
[3]Department of Community Health and Behavioral Sciences, School of Public Health, College of Health Sciences, Makerere University, Kampala, Uganda
[4]Department of Epidemiology and Biostatistics, School of Public Health, College of Health Sciences, Makerere University, Kampala, Uganda
[5]Department of Health Policy, Planning and Management, School of Public Health, College of Health Sciences, Makerere University, Kampala, Uganda
[6]Bill & Melinda Gates Foundation, Seattle, Washington, USA

**Acknowledgements** The authors wish to thank the study participants who provided the information that led to this publication. The research assistants are also appreciated for their contribution to this study.

**Contributors** RN, NC, SNKa, AN, WS, LLT and RKW conceptualised and designed the study. RN, NC, SNKa, AN, IW, SK supported the data collection. RN, NC, SNKa, AN, STW, IW, SK, SNKi, WS, LLT and RKW contributed to analysis and interpretation of findings. RN, NC, STW wrote the first draft of the manuscript. SNKa, AN, IW, SK, SNKi, WS, LLT and RKW critically reviewed the draft manuscript. All authors read and approved the final manuscript. RN is acting as the guarantor for this work.

**Funding** This work was supported in whole or in part, by the Bill & Melinda Gates Foundation [Opportunity ID: INV-019313]. The views, opinions and content of this publication are those of the authors and do not necessarily reflect the views, opinions, or policies of the Bill and Melinda Gates Foundation.

**Competing interests** None declared.

**Patient and public involvement** Patients and/or the public were not involved in the design, or conduct, or reporting, or dissemination plans of this research.

**Patient consent for publication** Not applicable.

**Ethics approval** This study involves human participants and was approved by Makerere University School of Public Health Higher Degrees Research and Ethics Committee (protocol SPH-2021-150). Participants gave informed consent to participate in the study before taking part.

**Provenance and peer review** Not commissioned; externally peer reviewed.

**Data availability statement** Data are available upon reasonable request. The data are available from the corresponding author on reasonable request.

**ORCID iDs**
Rawlance Ndejjo http://orcid.org/0000-0001-9263-557X
Nuole Chen http://orcid.org/0000-0003-0700-3784
Steven N Kabwama http://orcid.org/0000-0002-0354-9571
Solomon Tsebeni Wafula http://orcid.org/0000-0002-6405-015X
Lily L Tsai http://orcid.org/0000-0002-5264-4655

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
