## [Reviewer comments · BMJ Open]

ARTICLE DETAILS

TITLE (PROVISIONAL)	Uptake of COVID-19 vaccines and associated factors among adults in Uganda: a cross-sectional survey
AUTHORS	Ndejjo, Rawlance; Chen, Nuole; Kabwama, Steven; Namale, Alice; Wafula, Solomon; Wanyana, Irene; Kizito, Susan; Kiwanuka, Suzanne; Sambisa, William; Tsai, Lily; Wanyenze, Rhoda K

VERSION 1 – REVIEW

REVIEWER	Olomofe, Charles O. East Tennessee State University, Biostatistics and Epidemiology
REVIEW RETURNED	21-Oct-2022

GENERAL COMMENTS	This is research is well-presented. I left some comments on the methodology and ethics in the attached PDF file with sticky notes. - The reviewer provided a marked copy with additional comments. Please contact the publisher for full details.
--

REVIEWER	Sallam, Malik The University of Jordan
REVIEW RETURNED	18-Nov-2022

GENERAL COMMENTS	Thanks for the invitation to review this manuscript. In the current study, Rawlance Ndejjo et al. investigated COVID-19 vaccine uptake and its determinants among a sample of adult Ugandans. The authors utilized a phone-based survey to investigate the relevant aim of understanding the factors behind low coverage of COVID-19 vaccination in the country. The study results pointed to vaccine unavailability as the major factor behind being non-vaccinated and also the results pointed to low prevalence of COVID-19 vaccine hesitancy in the Eastern African country. Overall, the manuscript is well written with a clear description of the methods and results and with valid design. Importantly, the authors elaborated well on the potential limitations. The study also followed the ethical guidelines and disclosed the potential conflicts of interest. Therefore, I endorse the manuscript for publication following a minor revision: Abstract: Please add a statement to display the numbers and percentages of participants who received the primary dose, incomplete primary dose, did not receive any doses but willing to get vaccinated, did not receive any doses but hesitant to get vaccinated, and did not receive any doses and resist to get vaccinated.
---

	Introduction: Well written. However, I would suggest adding a few paragraphs to introduce the reader into the following themes:  1. COVID-19 vaccine hesitancy was relatively low in Uganda: https://doi.org/10.2147%2FJMDH.S347669 2. Type of COVID-19 vaccines used in Uganda: https://covid19.trackvaccines.org/country/uganda/ Methods: The sampling approach and study design were robust. The methods were described clearly to allow replication of the study. Sample size was sufficient although the design based on mobile phone survey might result in selection bias; however, the authors elaborated on this potential caveat. I have only a minor comment regarding the methods:  1. Did the researchers offer any incentives for participation? This can result in selection bias towards individuals with a lower socio-economic status. Results: Described clearly. One minor comment regarding Figure 1:  1. Please add the numbers to the figure. The authors can also consider stratifying the vaccine preference based on uptake/willingness to get vaccinated. Discussion: Well written with proper referencing. One minor comment:  1. Please elaborate on booster dose coverage in Uganda. Thank you!
--	--

VERSION 1 – AUTHOR RESPONSE

Reviewer: 1

Comments to the Author:

Comment 1: This research is well-presented.

Response: Thank you for the compliment.

Comment 2: I think there is need to be more explicit about the sampling strategy. The quota sampling should be more detailed than this.

Response: We have added information regarding the distribution used in setting quotas as below. We also referenced an earlier article published in the same journal with further details on the sampling. The distribution of cases at the time was as follows: age: 18–35 years (51%), 36–55 years (37%), 56–65 years (8%), 65+ years (4%); sex: male (60%) and female (40%); and region: central (55%) and 15% for each of Eastern, Western, and Northern regions. (Page 6, Line 124 – 127)

Comment 3: Were the questionnaires back-translated from these languages to English?

Response: Our quality control mechanism was to validate the questionnaire translations by independent translators following the Translation, Review, Adjudication, Pretesting, and Documentation (TRAPD) team approach (<https://ccsg.isr.umich.edu/chapters/translation/overview/>) as this just as effective with a shorter turnaround time.

Comment 4: On the average, how long does each call last? It might also be interesting to know the cost implication of these calls.

Response: The average interview time was 26 minutes. (Page 7, Line 147). Respondents did not have to incur any costs for the phone interview.

Comment 5: Was the verbal consent or phone interview recorded? Or how does the Ethics Review Committee ascertain consent was granted by participants?

Response: The consent form was read to each participant at the beginning of the interview and their consent was recorded on the mobile device to activate the survey questionnaire. The research team was trained in ethics and administering consent and was supervised during data collection.

Comment 6: Were the respondents compensated in any way for their time?

Response: Yes, respondents were provided phone credit of 1.5 US dollars to compensate for their time. (Page 21, Line 366-367)

Comment 7: 94% response rate for phone survey is remarkable!

Response: We used a number of strategies to ensure a high response rate. First, we were following up previous survey participants. Second, the research team was flexible to make interview appointments with respondents at their convenient time. Thirdly, we compensated participants for their time which may also have encouraged participation. We have added these explanations to share our experience. (Page 21, Line 364-367) As a side note, we followed up the same participants for a third survey in June 2022 and the response rate was still over 90%.

Reviewer: 2

Comment 1: In the current study, Rawlance Ndejjo et al. investigated COVID-19 vaccine uptake and its determinants among a sample of adult Ugandans. The authors utilized a phone-based survey to investigate the relevant aim of understanding the factors behind low coverage of COVID-19 vaccination in the country.

The study results pointed to vaccine unavailability as the major factor behind being non-vaccinated and also the results pointed to low prevalence of COVID-19 vaccine hesitancy in the Eastern African country.

Overall, the manuscript is well written with a clear description of the methods and results and with valid design. Importantly, the authors elaborated well on the potential limitations.

The study also followed the ethical guidelines and disclosed the potential conflicts of interest.

Therefore, I endorse the manuscript for publication following a minor revision:

Response: Thank you for the summary of the manuscript.

Comment 2: Abstract: Please add a statement to display the numbers and percentages of participants who received the primary dose, incomplete primary dose, did not receive any doses but willing to get vaccinated, did not receive any doses but hesitant to get vaccinated, and did not receive any doses and resist to get vaccinated.

Response: Unfortunately, due to the abstract word count of 300 words (which we have fully used up), we are unable to add all the important details but we believe that by having the total number of respondents, these numbers can be easily estimated.

Comment 3: Introduction: Well written. However, I would suggest adding a few paragraphs to introduce the reader into the following themes:

1. COVID-19 vaccine hesitancy was relatively low in Uganda:

<https://doi.org/10.2147%2FJMDH.S347669>

2. Type of COVID-19 vaccines used in Uganda: <https://covid19.trackvaccines.org/country/uganda/>

Response: Thank you for pointing us to this important literature. We have added a sentence on the low vaccine hesitancy in Uganda and the appropriate references to the discussion. (Page 19, Line 303 – 304) Regarding the vaccine types, the information on the webpage is not up to date and we have opted not to reference it.

Methods: The sampling approach and study design were robust. The methods were described clearly to allow replication of the study. Sample size was sufficient although the design based on mobile phone survey might result in selection bias; however, the authors elaborated on this potential caveat. I have only a minor comment regarding the methods:

Comment 4: Did the researchers offer any incentives for participation? This can result in selection bias towards individuals with a lower socio-economic status.

Response: We provided phone credit of 1.5 US dollars (determined together with the ethics committee) to compensate for the respondent's time (see p. 21, 376-377). Due to the high response rate recorded, we believe there was low selection bias due to socio-economic status. We note that sampling was informed by the quotas on age, sex, and region.

Comment 5: Results: Described clearly. One minor comment regarding Figure 1:
We have addressed the comment on Figure 1.

Comment 6: Please add the numbers to the figure. The authors can also consider stratifying the vaccine preference based on uptake/willingness to get vaccinated.

Response: We have added numbers to the figures as requested. As the figure is showing willingness to obtain a given vaccine type, it also reflects willingness to get vaccinated. Moreover, with the very high intention to vaccinate recorded in the survey, less variation would be expected in the stratification of this figure.

Comment 7. Discussion: Well written with proper referencing. One minor comment: Please elaborate on booster dose coverage in Uganda.

Response: Uganda had not begun giving out booster doses by the time of this survey (November 2021). Booster doses were only available from 2022.

Thank you.

VERSION 2 – REVIEW

REVIEWER	Olomofe, Charles O. East Tennessee State University, Biostatistics and Epidemiology
REVIEW RETURNED	23-Jan-2023
GENERAL COMMENTS	None